# Influence of Dilution on the Mechanical Properties and Microstructure of Polyurethane-Cement Based Composite Surface Coating

**DOI:** 10.3390/polym16010146

**Published:** 2024-01-03

**Authors:** Chao Xie, Yufeng Shi, Ping Wu, Binqiang Sun, Yaqiang Yue

**Affiliations:** 1Civil Engineering Department, Lanzhou Jiaotong University, Lanzhou 730070, China; shiyf25260@163.com (Y.S.); 14709328852@163.com (Y.Y.); 2CSCEC AECOM Consultants Co., Ltd., Lanzhou 730070, China; y252666@163.com; 3Gansu Yi’an Construction Technology Group Co., Ltd., Lanzhou 730070, China; sunbq8882023@163.com

**Keywords:** polyurethane-cement based composite, coating, mechanical properties, microstructure, positron annihilation lifetime spectroscopy, grey relation theory

## Abstract

Polyurethane-cement composite are widely used in modern civil engineering, and the method of adding diluent is often used to adjust the construction process to adapt to the engineering environment. Studies have shown that the addition of diluent impacts the performance of polyurethane-cement based composite surface coatings, but there have been few reports on the influence of diluent content on the mechanical properties and microstructure of the coatings. To address this, polyurethane coatings with different diluent contents were prepared, and positron annihilation lifetime spectroscopy was used to test the microstructure of the coatings. The tensile strength and elongation at rupture were tested using a universal material testing machine, and the fracture interface morphology of each coating was observed by scanning electron microscopy. Finally, the correlation between the microstructure parameters and the mechanical properties of the coating was analyzed using grey relation theory. The results demonstrated that with the increase in diluent content, (i) the average radius of the free volume hole (*R*) and the free volume fraction (*F_V_*) of the coating both showed a trend of first decreasing and then increasing. The value of R was between 3.04 and 3.24 Å, and the value of *F_V_* was between 2.08 and 2.84%. (ii) The tensile strength of the coating increased first and then decreased, while the elongation at rupture decreased first and then increased. Among them, the value of tensile strength was between 3.23 and 4.02 MPa, and the value of elongation at fracture was between 49.34 and 63.04%. In addition, the free volume in polymers plays a crucial role in facilitating the migration of molecular chain segments and is closely related to the macroscopic mechanical properties of polymers. A correlation analysis showed that the *R* value of the coating had the greatest influence on its tensile strength, while *F_V_* showed a higher correlation with the elongation at rupture.

## 1. Aims and Scopes

The aim of this paper is to study the effect of diluent content on the mechanical properties of a polyurethane-cement based composite surface coating and to take the free volume as the evaluation index of the microstructure change in the coating, combined with the study of macroscopic mechanical properties, to further analyze its change rule and development mechanism. Therefore, the research results of this paper have guiding significance for the construction of polyurethane-cement based composite surface coatings in practical engineering.

The research in this paper involves polymer analysis and characterization, polymer composites, and polymer films. The paint used in this paper is a one-component wet-curable polyurethane coating. Five groups of polyurethane coatings with different diluent contents have been designed as test specimens to carry out micro- and macro-tests. The test process is described in detail in Section 3 of this paper.

## 2. Introduction

Many resources are used every year to repair the structures associated with transportation infrastructure because of a lack of durability in the concrete used in their construction [1]. Studies have found that the durability of the structure can be effectively improved by applying a coating to the surface of the structure as a protective layer [2,3,4]. Therefore, composite materials composed of polymer- and cement-based materials are widely used in the construction of transportation facilities [5], and polyurethane has been widely used for surface coating parts of a polyurethane-cement based composite material due to its excellent resistance to weather, water, and abrasion and its flexibility at low temperatures [6,7,8,9]. The tensile strength, toughness (elongation at rupture), and other mechanical properties of the coating are related to the cracking, spalling, and other degradation phenomena occurring in the application process [10,11,12,13], thus they have a significant impact on the protective effect of the coating [14,15,16]. In view of the above, studying the mechanical properties of a coating is crucial in the predicting its performance upon application [17].

Jiang et al. [18] studied the mechanical properties of polyurethane coatings after introducing multiwalled carbon nanotubes and determined that in situ polymerization was better than blending polymerization in improving the mechanical properties of polyurethane coatings. Other research [19,20,21] has shown that the mechanical properties of polyurethane coatings are closely related to the phase separation and content ratio of the hard and soft segments. Gite et al. [22] found that, compared with the NCO/OH ratio of 1∶1, the tensile strength of the film with a NCO/OH ratio of 1.2:1 is increased by 1.60 to 6.00%, and the elongation at fracture is decreased by 1.22 to 13.08%. In addition, the tensile strength of the coating increased with an increase in hydroxyl content when the NCO:OH ratio remained unchanged. Li et al. [23] prepared a ZnO-nanoparticle-reinforced polyurethane coating by solution blending and found that the tensile strength of the coating showed a trend of first increasing and then decreasing as the ZnO content was increased, with the opposite trend observed for the fracture elongation. The value of the tensile strength was between 7.24 and 17.83 MPa, and the value of the elongation at fracture was between 54.79 and 98.37%.

The aforementioned studies explored the effects of various factors on the mechanical properties of polyurethane coatings from different perspectives. In practical engineering, the fluidity of the paint is usually adjusted by adding diluent to meet the needs of the project [24,25], and the addition of diluent has a greater impact on the performance of the cured coating [26,27]. However, there is little published research on the influence of diluent content on the mechanical properties of coatings. This forms the main objective of the present study.

In order to clarify the mechanism of the effect of diluent on the mechanical properties of the coating, the free volume will be used as the evaluation index of the microstructure change in the coating and linked to the study of macroscopic mechanical properties in order to analyze its variation rule [28]. The free volume refers to the volume not occupied by molecules in the polymer, which is dispersed in the whole material in the form of holes. It is an intrinsic defect in polymer materials and provides the space needed for molecular chain movement. Therefore, the mechanical properties of a polymer are closely related to its free volume [29]. Positron annihilation lifetime spectroscopy (PALS) is an important method for measuring the free volume of a polymer [30,31,32], and the polymer free volume information can be comprehensively understood by analyzing the *o*-Ps annihilation lifetime (*τ*_3_) and intensity (*I*_3_) [33,34,35].

To sum up, the study described herein designed paint systems with different diluent contents and prepared corresponding polyurethane coatings. The mechanical properties and fracture morphologies of the coatings were investigated using a material testing machine and scanning electron microscope, and the free volume of each component of the coatings was measured by PALS. Variations in the mechanical parameters of the coatings, such as tensile strength and elongation at rupture, were analyzed from a microscopic perspective. At the same time, the relationship between the free volume indexes and the mechanical properties of each coating was quantitatively determined by grey relation theory. Finally, the influence of mechanical properties and the microstructure of polyurethane-cement based composite surface coatings and their development mechanisms are obtained.

## 3. Experimental

### 3.1. Materials

The polyurethane paint and diluent used in this paper were provided by Urumqi You Bao Te Anti-corrosion Paint Co. Ltd. (Urumqi, China). The polyurethane paint was a one-component moisture-curing paint, and the diluent was nonactive diluent matched with polyurethane anticorrosive paint. The basic properties of the polyurethane paint are shown in Table 1.

### 3.2. Sample Preparation

To prepare the experimental paint, diluents were added at 10%, 15%, 20%, 25%, and 30% of the mass of polyurethane paint. Then, the coatings were spread on a template, cured for 12 h at room temperature (20 °C ± 2 °C, 55% ± 3% humidity), and put into a coating curing instrument (SHBY-40b curing box, Beijing Aerospace Huayu Test Instrument Co. Ltd., Beijing, China). After curing for 7 d in the standard curing environment (temperature = 23 °C ± 2 °C, humidity = 50% ± 10%), the coating was taken out and further cured for 7 d at room temperature before its performance was tested. From here on, the coating samples with 10%, 15%, 20%, 25%, and 30% of diluent after curing are denoted as S-10, S-15, S-20, S-25, and S-30, respectively.

### 3.3. Mechanical Properties Tests

The mechanical properties of the prepared coatings were measured using a universal material testing machine (WDW-50, EST Testing Instruments Inc., Shanghai, China). The crosshead speed was 50 mm/min during the tests. The coatings were cut into dumbbell shapes, and the mechanical properties were measured according to ASTM D638-14 specifications [36]. The tensile strength test of the process is shown in Figure 1.

### 3.4. Positron Annihilation Lifetime Spectroscopy (PALS)

The microstructural information of the coating was tested using PALS, with the experiments carried out on the positron research platform of the Institute of High Energy Physics at the Chinese Academy of Sciences. A ^22^Na radioactive source was adopted as the positron source, and the strength was about 13 μCi. Two identical samples were clamped to either side of the radioactive source, forming a typical “sandwich” structure, as shown in Figure 2. A pair of BaF_2_ scintillator detectors were employed in the positron annihilation lifetime spectrometer to detect gamma rays released after positron annihilation, as shown in Figure 3. The positron annihilation lifetime spectrum was measured by fast-slow coincidence measurement technology, and the time resolution of the spectrometer was about 195 ps. The lifetime spectrum was measured with two million counts to ensure statistical accuracy. The electronic plug-in of the measurement system was a standard NIM made by EG&G Inc., and the LT9.0 program was used to analyze the positron lifetime spectra [37].

### 3.5. Scanning Electron Microscopy (SEM)

The morphologies of the fractured interfaces of the coatings were obtained from tensile testing using a scanning electron microscope (Versa 3D, FEI Co., Hillsboro, OR, USA). The samples were installed on the section sample table with the fracture facing up and “bypass” treatment performed using conductive tape. Then, the surfaces of the samples were made conductive via ion sputtering with a sputtering time of 60 s. After the sample was processed, the high vacuum mode was selected with a load voltage of 5 kV and the working distance was adjusted to 10 mm while maintaining the above instrument parameters. The coating microform was then analyzed under different magnifications.

## 4. Results and Discussion

### 4.1. Free Volume

The microstructure of the sample was tested by PALS and the free volume information of the sample was calculated based on the test results. This was used as a parameter index to investigate the microstructure of the coating [38]. The positron annihilation lifetime spectrum was resolved into three lifetime components: *τ*_1_, *τ*_2_, and *τ*_3_, corresponding to the *p*-Ps annihilation lifetime, free positron annihilation lifetime, and *o*-Ps annihilation lifetime, respectively [39]. Among them, *τ*_3_ originates from *o*-Ps annihilation in polymer free volume holes, which can be localized preferentially in the cavities. Therefore, information on the holes and defects in the samples can be obtained by analyzing the annihilation of *o*-Ps [40]. This study focused on investigating the variation in the *o*-Ps annihilation lifetime of each sample.

The PALS spectra obtained for the samples are shown in Figure 4. They were analyzed and fitted using the LT9.0 program [37], and the obtained values for *τ*_3_, *I*_3_, and the fitting factor are provided in Figure 5 and Figure 6. The fitting factor represents the degree of fitting, and the closer it is to 1, the closer the fitting result is to the actual measured value.

The *τ*_3_ and *I*_3_ of the coating initially decreased and then increased as the diluent content in the paint was increased, as shown in Figure 6, where *τ*_3_ and *I*_3_ represent the aperture size and total content of free volume, respectively. To quantitatively investigate the change in free volume and further clarify the observed changes in the coating microstructure, Equations (1)–(3) were used to calculate the average radius of the free volume hole (*R*) and the free volume fraction (*F_V_*). Therein, *R* is calculated by Equation (1) [41,42].
(1)$$\tau_{3}=\frac{1}{2}{\left[1-\left(\frac{R}{R_{0}}\right)+\left(\frac{1}{2\pi }\right)\sin\left(\frac{2\pi R}{R_{0}}\right)\right]}^{-1}$$

The units of *τ*_3_ and R were ns and Å, respectively. *R*_0_ = *R* + Δ*R* and the experimental value for Δ*R* was 1.66 Å [43,44], which is the empirical electron layer thickness. *F_V_* is calculated from Equation (2):(2)$$F_{V}=CV_{f}I_{3}$$

C is a structure constant and its empirical value is 0.0018 [28,43,44]. *I*_3_ is the annihilation intensity of *o*-Ps, which was evaluated according to Figure 6. *V_f_* is the average hole volume of the free volume, which was calculated according to Equation (3):(3)$$V_{f}=\frac{4}{3}\pi R^{3}$$

The free volume parameters of each sample were calculated using the above calculation method and are shown in Figure 6.

From Figure 7, it can be seen that when the diluent content was increased, both the *R* and *F_V_* of the coating exhibited a trend of initially decreasing and then increasing, but at different rates. In comparison to S-10, the diluent content of S-15, S-20, S-25, and S-30 increased by, respectively, 5%, 10%, 15%, and 20%; *R* decreased by, respectively, 0.9%, 6.2%, 2.5%, and 1.9%; and *F_V_* decreased by, respectively, 5.5%, 26.8%, 16.0%, and 11.1%.

The above two free-volume parameters of S-20 are the smallest, which indicates that its microscopic defect is the smallest among the five groups of samples, no matter the size of a single free-volume hole or the free volume content, and the whole structure is the most compact. It also means that the activity space of the molecular chain segment of S-20 was the smallest, and, therefore, the mobility of the chain segment and the deformation capacity of the sample were the lowest. However, the *R* and *F_V_* of S-10 are the maximum values of all the samples studied, indicating that the mobility of the chain segment and the deformation capacity of S-10 were the strongest [28].

### 4.2. Mechanical Properties

The test results of the tensile strength and elongation at rupture of each group are shown in Figure 8. The tensile strength was 3.23–4.02 MPa and the elongation at rupture was 49.34–63.04%, which are similar to data reported previously and with similar ranges in variation [22,45,46].

The tensile strength of the coating exhibits an initial increase followed by a decrease with increasing diluent content, as depicted in Figure 8. Conversely, the elongation at break demonstrates an initial decrease followed by an increase. Among the five groups of specimens investigated in this study, S-20 displays the highest tensile strength, while S-10 exhibits the lowest. Specifically, the tensile strengths of S-15, S-20, S-25, and S-30 are 1.11 times, 1.24 times, 1.19 times, and 1.14 times that of S-10, respectively. On the other hand, S-10 shows the highest elongation at break, whereas it is the lowest for S-20. The elongation at break values for S-15, S-20, S-25, and S-30 are 99.4%, 78.3%, 86.6%, and 93.3%, respectively, compared to that of S-10.

Stress–strain curves for all the samples studied are shown in Figure 9, where it can be seen that different diluent contents impact the mechanical response of the coating under external load. The stress–strain curve for S-10 was somewhat more complex, where the stress basically remained unchanged and the strain continued to increase after the curve reached point A. It then rose further from point B, which may have been caused by orientation hardening. Subsequently, starting from point C, the upward slope of the curve gradually flattened out. When it reached the highest point, D, the stress dropped sharply and the strain increased slightly before a fracture occurred. The variation in the stress–strain curve for S-15 is similar to that of S-10. First, it rose to point A with a steep slope, followed by a gentle curve from point A to point B. Subsequently, it continued to rise and reached its peak at point D. However, different from S-10, the stress slightly decreased with increasing strain after reaching its maximum threshold until failure occurred. The stress–strain curve for S-20 is relatively simple. The slope of the curve was relatively steep at the initial stage of loading, and then it gradually decreased from point A. When the stress value reached its maximum, it dropped slightly and then broke. There is no obvious yield point during the whole loading process. The stress–strain curve of S-25 is basically the same as that of S-20, but the strain increased from the highest stress point to the breaking point of the coating. The stress–strain curve showed obvious piecewise changes starting from S-30. At the initial stage of loading, the stress–strain curve for S-30 was similar to those for S-20 and S-25 but with a steeper slope. However, when it reached the highest point, A, the curve flattened out and the stress essentially remained unchanged, but the strain continued to increase. After that, the stress dropped sharply at point B and broke at point C.

The above analysis reveals that S-20 exhibits low toughness, with fractures occurring without significant plastic deformation throughout the loading process and with a corresponding low fracture elongation. In contrast, for S-10 to S-30, there is an initial decrease followed by an increase in plastic deformation, fracture elongation, and toughness.

The brittle fracture is caused by the tensile component of the applied stress, which leads to a relatively flat fracture surface. On the other hand, the ductile fracture is caused by the shear component, resulting in a relatively rough topography on the fracture surface. The actual ratio of the tensile and shear components during the fracture process depends on both the stress system and specimen performance [47,48,49]. As depicted in Figure 10, S-20 displays a smooth and flat fracture surface with neatly aligned cracks, resembling glass-like characteristics indicative of the brittle fracture. In contrast, the S-10 displays a significantly irregular shape with pronounced epitaxial deformation and cell structure observed in cross sections of the S-10, S-15, and S-30 coatings. This suggests the evident occurrence of plastic deformation during stress-induced fractures. Additionally, dimple-like structures are also present in S-10 and S-15 due to localized matrix yielding [50,51], indicating clear signs of ductile fracturing behavior. The roughness of the fracture surface of S-25 lies between that of the S-20 and S-30 coatings, while exhibiting an intermediate failure state between brittle and ductile fractures. Consequently, as the diluent content increases, there is a transition from ductile fractures to brittle fractures, followed by ductile fractures.

From the above, it is clear that the content of diluent in the paint has a significant impact on the mechanical properties of the cured coating because of the resulting change in the microstructure of the coating caused by the diluent [28,29,38]. The results in Section 4.1 demonstrate a non-linear relationship between the diluent content in the paint and in both *R* and *F_V_*. Specifically, *R* and *F_V_* initially decrease before reaching a turning point and increasing. The presence of a larger mean pore diameter and a higher free volume fraction in the coating facilitates the migration of molecular chain segments and enhances the flexibility of side chain rotation, thereby promoting the increased mobility of molecular chain segments under external loads and resulting in enhanced fracture elongation. In addition, the cellular structure generated during the deformation process can absorb or dissipate energy to avoid the instantaneous rupture caused by the accumulation of energy in the material. Meanwhile, the local plastic deformation of the matrix further improves its toughness [52].

However, with an increase in free volume aperture size and content, the limiting capacity of the matrix to its molecular chain segment movement will be reduced [23], and the load transfer efficiency will be reduced, thus leading to a decrease in the tensile strength. In addition, the material fracture commonly occurs in areas with many material defects, and large void defects will lead to a concentration of stress in the specimen, further reducing the tensile strength of the material. Therefore, as the *R* and *F_V_* of the paint increase, the tensile strength of the coating decreases.

### 4.3. Correlation Analysis

As can be seen from the above analysis and Figure 11, the tensile strength of the coating was negatively correlated with its *R* and *F_V_*, and the elongation at rupture of the coating was positively correlated with these variables. In addition, both offered a good correlation, with the coefficient of determination (*R*-squared) for linear models reaching above 0.90. To more clearly define the degree of correlation between the microstructure parameters and the mechanical properties of the coating, quantitative analysis was performed using grey relation theory [53].

The grey relation analysis method can use limited data to obtain the degree of influence among many factors of the system or the degree of contribution of several subfactors (subsequences) to the main factor (parent sequences). Therefore, the mapping variables (parent sequences) representing the behavior characteristics of the system were represented by the tensile strength and elongation at rupture of the coating and were denoted as X_T_ and X_E_, respectively. The *R* and *F_V_* of the coating were taken as the effective factors (subsequences) affecting the main behavior of the system, which were denoted as X_R_ and X_F_, respectively. The correlation degree was calculated as follows:

First, the initial value image for each sequence was calculated, with the results provided in Table 2.

Second, the difference sequence between each subsequence and parent sequence was calculated, with the results provided in Table 3.

Third, the correlation coefficient was calculated, and the data from Table 4 were used in Equation (4), with ξ selected as 0.5.
(4)$$\gamma_{0i}(k)=[\Delta (min)+\xi \Delta (max)]⁄[\Delta_{i}\left(k\right)+\xi \Delta (max)]$$

The correlation coefficient between each subsequence and the parent sequence was thus obtained. These are provided in Table 4.

Finally, the grey correlation degree was calculated and the data in Table 5 were used in Equation (5).
(5)$$\gamma_{0i}=\frac{1}{n}\sum_{k=1}^{n}\gamma_{0i}\left(k\right),i=1,\;2,\;3...m$$

The grey correlation degree between each subsequence and the parent sequence was obtained and these are shown in Table 5.

The calculation results showed that the correlations between the tensile strength of the coating and its *R* and *F_V_* were 0.6583 and 0.5728, respectively. This indicates that the *R* has a greater influence on the tensile strength of the coating among the microstructure parameters studied. The correlations between the elongation at rupture of the coating and its *R* and *F_V_* were 0.6690 and 0.7254, respectively. This indicates that the *F_V_*, i.e., the overall void content of the coating, has a higher correlation with its elongation at rupture. Therefore, the variation in the mechanical properties of different coatings can be predicted, to some extent, by comparing the corresponding microstructure parameters.

## 5. Conclusions

The microstructure, mechanical properties, and fracture interface morphologies of polyurethane coatings with different diluent contents were studied, and the correlation between the microstructure and mechanical properties of the coating was analyzed using grey relation theory. The following conclusions can be drawn:

(1) With an increase in diluent content in the polyurethane coating, the tensile strength of the coating initially increased and then decreased, and the value varied between 3.23 and 4.02 MPa; the elongation at rupture initially decreased and then increased, and the value varied between 49.34 and 63.04%. At the same time, the toughness of the coating initially decreased and then increased.

(2) Positron annihilation tests showed that the diluent for the polyurethane coating significantly influenced the microstructure of the cured coating, and the hole diameter and total content of free volume of the coating initially decreased and then increased as the diluent content was increased; the values are 3.04 to 3.24 Å and 2.08 to 2.84%, respectively.

(3) The mechanical properties of the coating are closely related to its microstructure. With an increase in the average radius of the free volume hole (*R*) and the free volume fraction (*F_V_*), the tensile strength of the coating decreased and the elongation at rupture increased. The correlation between the microstructure parameters of the coating and the mechanical properties of the coating was quantitatively analyzed using grey relation theory. It can be concluded that *R* has a greater influence on the tensile strength, while *F_V_* has a higher correlation with the elongation at rupture.

This study indicates that, when designing polyurethane-cement based composite for practical application, the properties of the surface coating for composites and the requirements for the paint in actual construction should be comprehensively considered to determine the most appropriate diluent content to guarantee a better use effect.

## Figures and Tables

**Figure 1 polymers-16-00146-f001:**
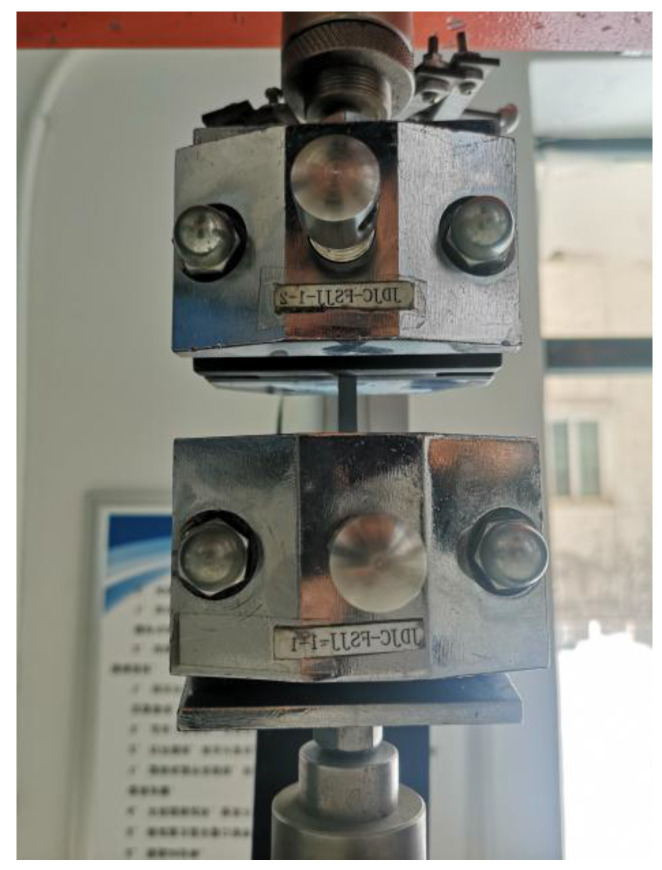
Tensile strength test.

**Figure 2 polymers-16-00146-f002:**
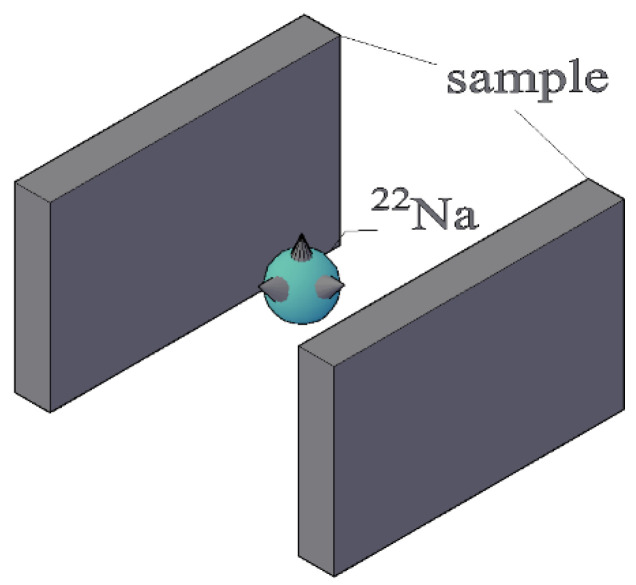
Sandwich structure formed by the clamps.

**Figure 3 polymers-16-00146-f003:**
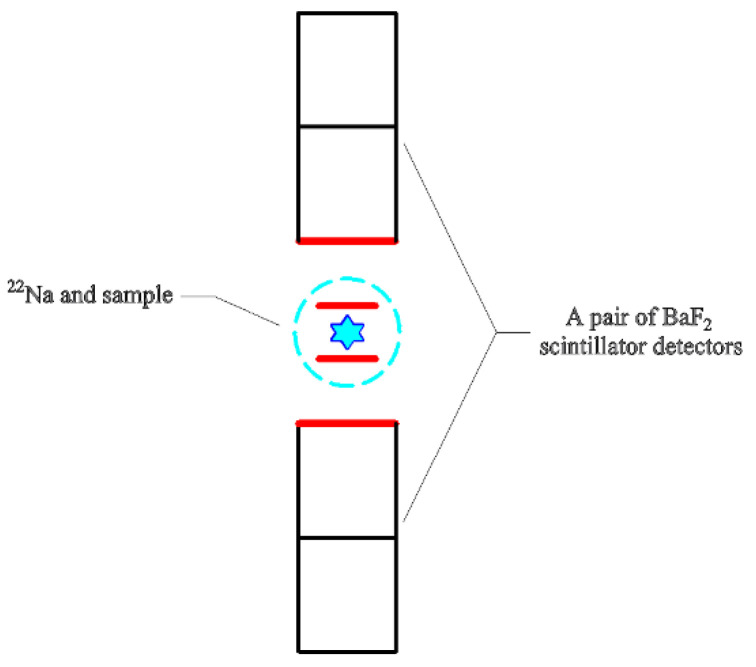
Positional relationship between the sample and scintillator detector.

**Figure 4 polymers-16-00146-f004:**
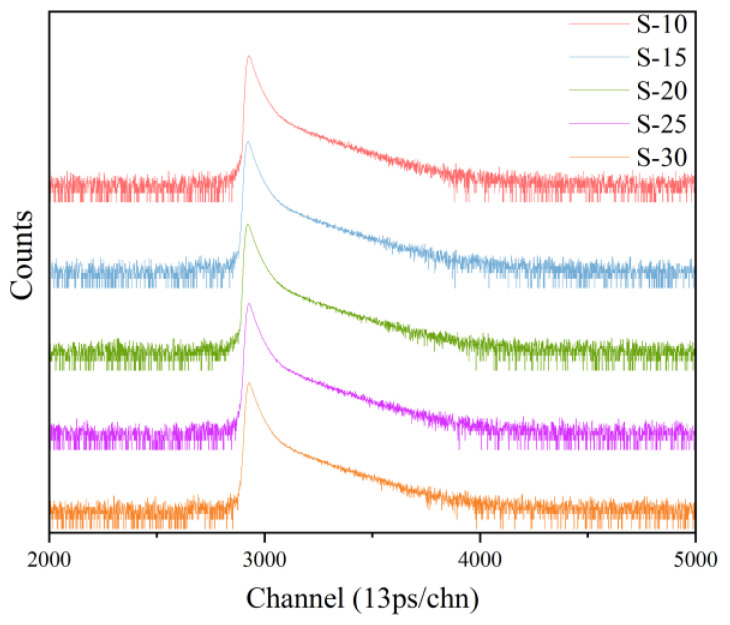
PALS spectrum of each sample.

**Figure 5 polymers-16-00146-f005:**
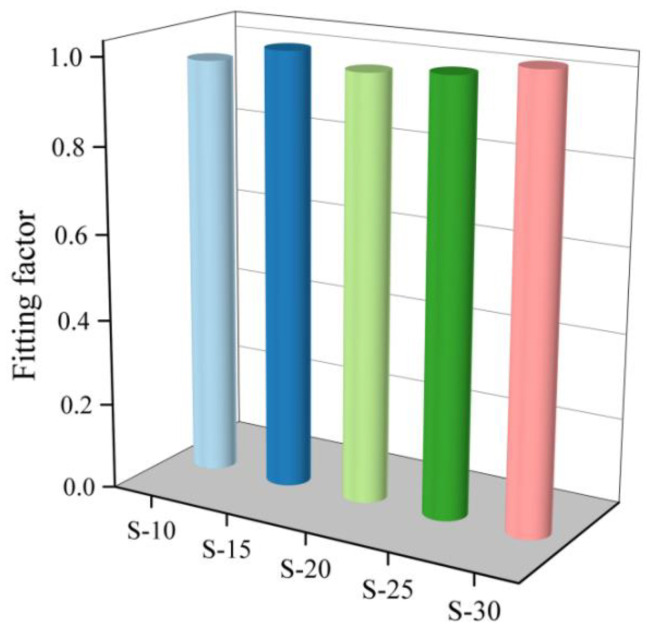
Each coating *o*-Ps annihilate spectrum fitting coefficient.

**Figure 6 polymers-16-00146-f006:**
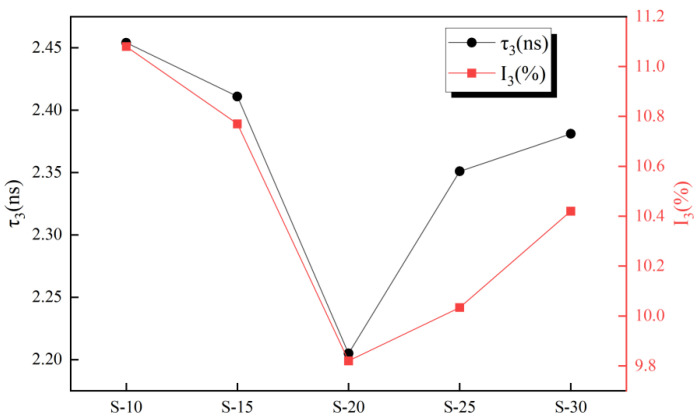
Each coating *o*-Ps annihilate life and annihilation intensity.

**Figure 7 polymers-16-00146-f007:**
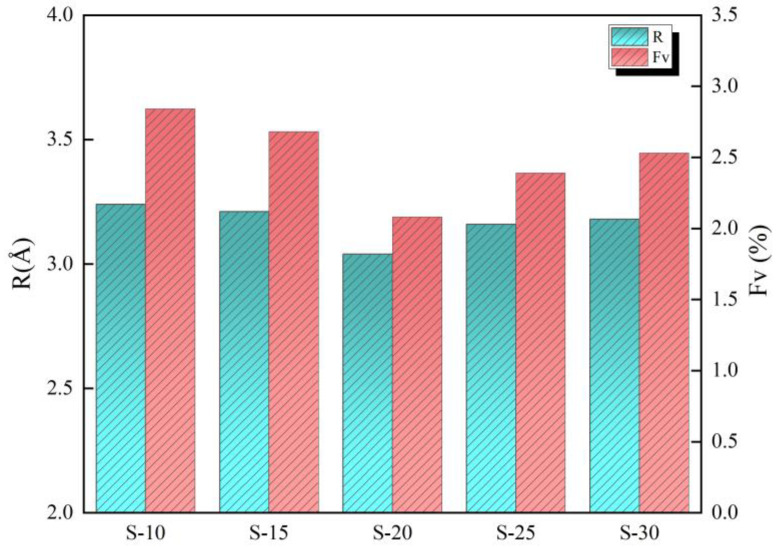
Results of free-volume parameters for samples.

**Figure 8 polymers-16-00146-f008:**
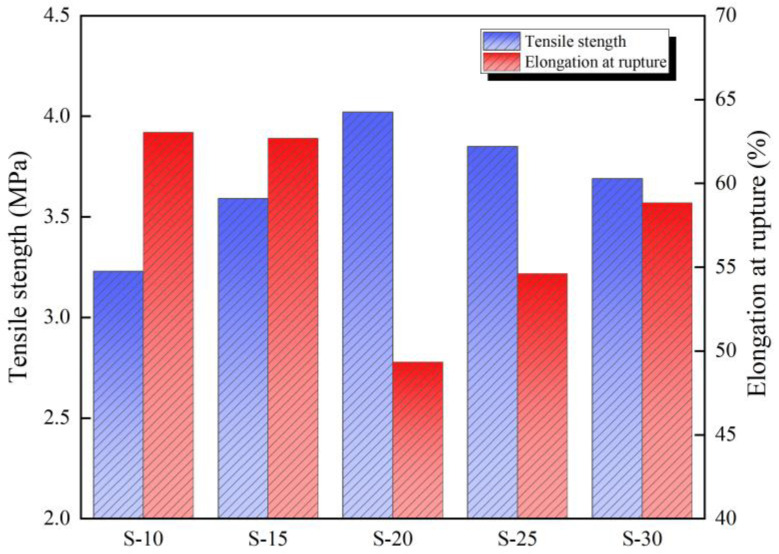
Test results of mechanical properties of each sample.

**Figure 9 polymers-16-00146-f009:**
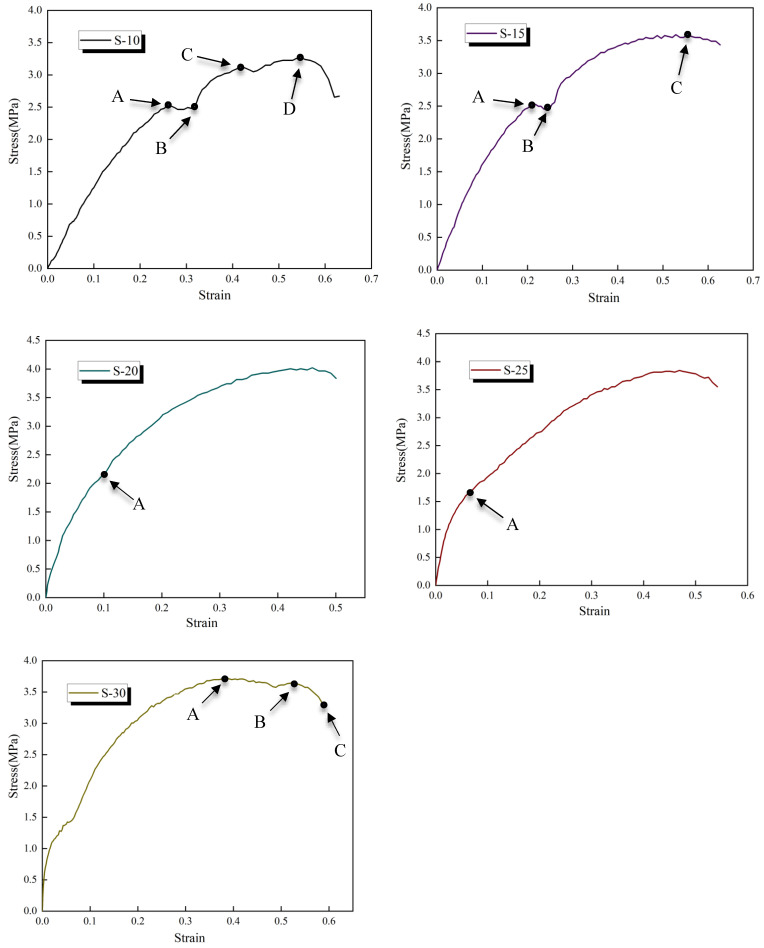
The stress–strain curve of each sample.

**Figure 10 polymers-16-00146-f010:**
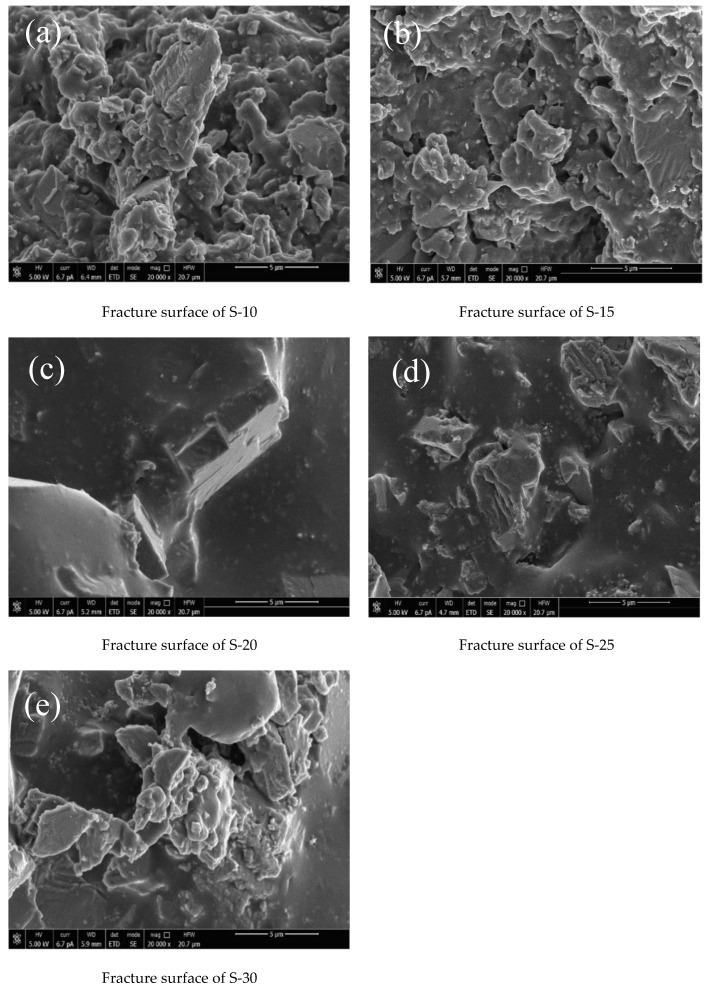
Cross-section SEM image of (**a**) S-10, (**b**) S-15, (**c**) S-20, (**d**) S-25, and (**e**) S-30.

**Figure 11 polymers-16-00146-f011:**
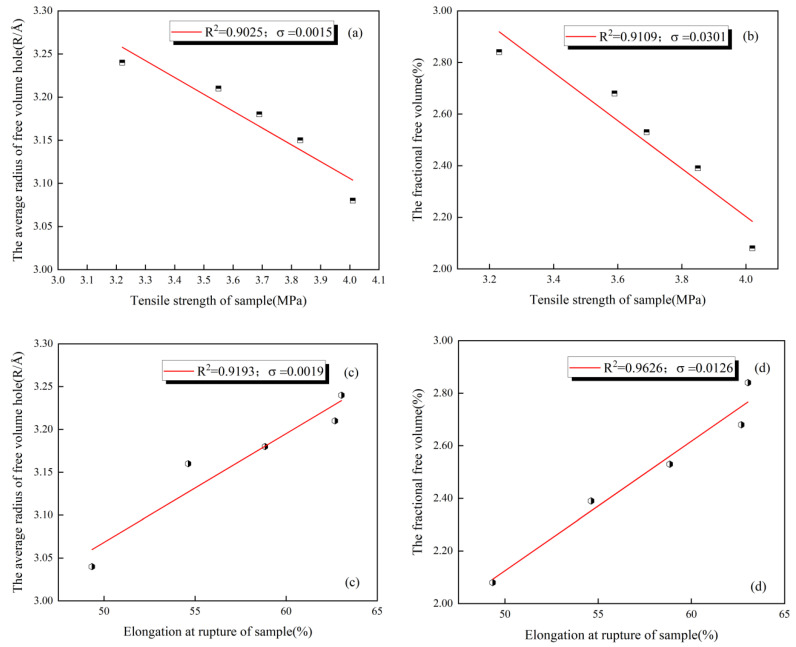
Correlation between mechanical properties and microstructure parameters of coating: (**a**) tensile strength and *R*, (**b**) tensile strength and *F_V_*, (**c**) elongation at rupture and *R*, and (**d**) elongation at rupture and *F_V_*.

**Table 1 polymers-16-00146-t001:** Basic properties of polyurethane paint.

Items	Results
Fineness/μm	40
No volatile matter content/%	67
Surface drying time/h	2
Practical drying time/h	20
Acid corrosion resistance (50 g/L H_2_SO_4_)	168 h (No abnormalities)
Alkali corrosion resistance (Saturated Ga(OH)_2_)	240 h (No abnormalities)

The above test temperature was 23 °C ± 2 °C and the humidity was 50% ± 5%.

**Table 2 polymers-16-00146-t002:** Initial value image for each sequence.

Sample	X_T_	X_E_	X_R_	X_F_
S-10	1	1	1	1
S-15	1.111455108	0.994289340	0.990740741	0.945270293
S-20	1.244582043	0.782677665	0.938271605	0.732078155
S-25	1.191950464	0.866434010	0.975308642	0.839822276
S-30	1.142414861	0.933375635	0.981481481	0.889148475

**Table 3 polymers-16-00146-t003:** Difference sequence between each subsequence and parent sequence.

Sample	Δ_T-R_	Δ_T-F_	Δ_E-R_	Δ_E-F_
S-10	0	0	0	0
S-15	0.120714368	0.166184815	0.003548599	0.049019047
S-20	0.306310438	0.512503889	0.155593940	0.050599510
S-25	0.216641822	0.352128189	0.108874632	0.026611734
S-30	0.160933379	0.253266386	0.048105847	0.044227160

**Table 4 polymers-16-00146-t004:** Correlation coefficient between each subsequence and parent sequence.

Sample	γ_T-R_(k)	γ_T-F_(k)	γ_E-R_(k)	γ_E-F_(k)
S-10	1	1	1	1
S-15	0.679772472	0.606602464	0.957526525	0.620063485
S-20	0.455506613	0.333331647	0.339567308	0.612559724
S-25	0.541878687	0.42120182	0.423561382	0.750386442
S-30	0.614238277	0.502927888	0.624483596	0.64398156

**Table 5 polymers-16-00146-t005:** Calculated degree of grey correlation.

γ_T-R_	γ_T-F_	γ_E-R_	γ_E-F_
0.6583	0.5728	0.6690	0.7254

## Data Availability

The data that support the findings of this study are available from the corresponding author upon reasonable request.

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
