# Peer review of "Influence of Dilution on the Mechanical Properties and Microstructure of Polyurethane-Cement Based Composite Surface Coating"

_polymers, 2024, doi:10.3390/polym16010146_

Round 1

Reviewer 1 Report

Comments and Suggestions for Authors

The article has an applied focus, and its results have undoubted practical significance. The deformation behavior of polymer-cement based composites is widely studied and in most cases predictable, but the authors apply relevant and exciting research methods, which adds relevance.

Nevertheless, I have a few suggestions to improve the quality of presentation of the results obtained:

1. The authors should pay more attention to the state of the art of the problem under study, since the vast majority of the cited sources cannot be called up-to-date.

2. The dependencies presented in Figures 6 and 7 definitely add beauty and attractiveness to the illustrations, but are extremely difficult to perceive because of the inverted coordinates. It is suggested to give them a more classical look.

3. Dependencies in Figures 5 and 10 do not have any confidence interval indicated. It is recommended to give the value of the standard deviation to make sure that the data obtained are not the result of experimental error.

4. Figure 8 requires a more thorough discussion. The choice of the position of points C (S-15), A (S-20 and S-25), B AND C (S-30) is non-obvious and seems subjective.

5. In addition to the SEM images in Figure 9, it is suggested to provide images taken at lower magnification to better represent the nature of the fracture surface.

Reviewer 2 Report

Comments and Suggestions for Authors

ID: Polymers-2709301

Title: Influence of dilution on the mechanical properties and microstructure of Polymer-cement based composite surface coating

In this research, the effect of dilution on the behavior of a polymer-cement based composite was investigated.

The addressed issue is important and the used test method sounds right. The manuscript has a good structure and it is well written. However, some minor issues need to be addressed.

1. Revise the abstract, important results should presented in numerical form in the abstract.

2. Avoid mass citations. for example 2~6, for a sentence that presents a basic knowledge.

3. The literature review should presented more comprehensively and in a better way. The chosen studies should be presented including this information 1: the aim of their study, 2: the material and used method, and 3: their main findings in numerical form.

4. Please add a subsection entitled “aim and scopes”.

5. Your aims and scopes should be revised. The significance of your study should be presented more clearly. For example, the total number of tests, or the number of mix designs.

6. From the title, I was expected to see this: you add different dosages of a dilator in a compound resulting from the mixture of a polymer and cement. As in the section.2 described you used commercial polyurethane paint from a factory. first, I suggest expressing the polymer type in the title, for example, “Influence of dilution on the mechanical properties and microstructure of polyurethane-cement based composite surface coating. second, you should provide more information on the used compounds, including the content of ingredients, and their mechanical and chemical properties.

7. Why you didn’t test the pure polyurethane paint as the control mixture?

8. Which standard was used for mechanical tests?

 9. As can be seen in Fig.7, the tensile strength and rapture strain have meaningful reverse relations. in this condition, the calculation of tensile energy is also appreciated.

10. Please add some pictures of the specimen during the tests.

11. More than describing the behaviors, you should explain the reasons. for example, why a step is seen in curves? you describe it as: “The tensile strength of the coating exhibits an initial increase followed by a decrease with increasing diluent content, as depicted in Fig. 7….” but no appreciable explanations were provided. also, if this behavior is seen in other studies, it is good to mention them.

12. On page 8 you mentioned that the fracture is brittle. while the tress-strain curves are indicating high ductility. I think you should revise the last paragraph of page 8. I cannot follow the explanations on how tensile-shear stresses influence the fracture surface morphology.

13. The title of each figure should be self-explanatory, especially figures such as Fig.9. please add short explanations for each a~e subfigure.

14. The results in the conclusion should presented in numerical and quantitive forms.

Comments on the Quality of English Language

the language is overally good, however it can be enhanced by a native english speaker.

Reviewer 3 Report

Comments and Suggestions for Authors

Two my mind, the work by Xie at al. has one main advantage: it is the use of positron annihilation lifetime spectroscopy for analysis of PU paint samples. This is a relatively rare method, the use of which with polymer objects can rarely be seen in work.

Therefore, the authors should explain why this method is better than more common ones such as X-ray diffraction.

In addition, the text of the manuscript contains microphotographs of samples of films of PU paints (Fig. 9), but the text does not contain a link to this figure and its description.

I would also like to note that the research is too applied in nature and I would like the authors to more strongly emphasize the fundamental nature of their work.

Round 2

Reviewer 2 Report

Comments and Suggestions for Authors

Thank you for revision. It is now suit for publication.

Reviewer 3 Report

Comments and Suggestions for Authors

The authors made required corrections.

Manucript can be accepted in the present form.

Comments on the Quality of English Language

 Minor editing of English language required